# Detection of Persistent Organic Pollutants in Omental Adipose Tissue from Patients with Diffuse-Gastric Cancer: A Pilot Study

**DOI:** 10.3390/cancers13194874

**Published:** 2021-09-29

**Authors:** Martine Perrot-Applanat, Cynthia Pimpie, German Cano-Sancho, Jean Philippe Antignac, Marc Pocard

**Affiliations:** 1INSERM U1275, Lariboisiere Hospital, University of Paris-Diderot-Paris7, F-75010 Paris, France; cynthia.pimpie@inserm.fr; 2INSERM U1275, CAP Paris-Tech, Lariboisière Hospital, 2 rue Ambroise Paré, F-75010 Paris, France; 3Oniris, INRAE, LABERCA, F-44307 Nantes, France; german.cano-sancho@oniris-nantes.fr (G.C.-S.); jean-philippe.antignac@oniris-nantes.fr (J.P.A.); 4Hepato-Biliary-Pancreatic Gastrointestinal Surgery and Liver Transplantation, Pitié-Salpêtrière Hospital, Assistance Publique/Hôpitaux de Paris, F-75013 Paris, France; marc.pocard@inserm.fr

**Keywords:** omental fat tissue, diffuse-gastric cancer (diffuse-GC), persistent organic pollutants (POPs), polychlorinated dibenzo-p-dioxins (PCDD), metastasis

## Abstract

**Simple Summary:**

This pilot study reported the observation that great omentum could be analyzed to detect persistent organic pollutants (POPs). Diffuse gastric cancer is an increasing disease that could be associated with pollutants’ exposition. Here, we report a specific POP profile regarding a patient not affected by cancer, nor by diffuse gastric cancer or other abdominal cancers. The widespread presence of a substantial list of POPs (PCDDs/Fs, PCBs, and brominated flame retardants) was found in the omentum from patients with diffuse gastric cancer with minor presence of some organochlorine pesticides.

**Abstract:**

The greater omentum represents a specific adipose tissue resected with gastric surgery for cancer. Diffuse gastric adenocarcinoma (diffuse-GC) is of major relevance among gastric cancers due to its unknown origin, aggressiveness, and metastasis in the peritoneal cavity. We postulated that persistent organic pollutants (POPs) could be detected in the greater omentum. Great omentum from patients with (i) diffuse-GC, or (ii) with other peritoneal metastatic cancer, and (iii) control group without cancer disease were analyzed for the distribution of a large panel of 96 POPs. POPs include polychlorinated dioxins/furans (PCDD/Fs), polychlorobiphenyls (PCBs), polybrominated diphenyl ethers (PBDE), polybrominated biphenyls (PBB), hexabromocyclododecanes, organochlorine pesticides, and polycyclic aromatic hydrocarbons (PAHs). The widespread presence of a substantial list of POPs (PCDDs/Fs, PCBs, and brominated flame retardants) was found in the omentum from patients with aggressive diffuse-GC, with minor presence of some organochlorine pesticides and PAHs at the low analyzed levels. Some chemicals appeared in larger concentrations in diffuse-GC or other cancer groups, including some PCDDs, PCB105, 123, 138, PBDE209, and PBB153. Overall, the present pilot study provides novel information regarding POPs levels in the omental fat, which is an understudied fat depot in terms of POPs load, and diffuse-GC association.

## 1. Introduction

Gastric cancer (GC) is a major health problem, the fourth among the most common cancers worldwide and the third in mortality rates [1,2]. The different sub-types include intestinal-, diffuse-, and mixed types according to the Lauren classification [3]. Unlike the decreasing incidence of the intestinal-type GC, the prevalence of the diffuse-type is increasing worldwide, especially in the USA and Europe [4]. Diffuse-type GC is a poorly differentiated, infiltrating, and scattered type cancer, and it is generally diagnosed at an advanced stage [5]. The implication of multiple signaling pathways has been identified in diffuse-GC [6]. Due to its aggressive behavior, diffuse-GC leads to a poor prognosis and treatment approaches are limited [7,8,9]. Identification of risk factors and clinically useful biomarkers remains an important goal in the management of early and late stages of diffuse-GC.

The major role of dietary, lifestyle, and environmental factors has been documented in the etiology of human cancers, including gastric cancers [10,11,12,13,14,15,16,17,18,19,20]. Exposure to environmental persistent organic pollutants (POPs) has also been pointed to as a risk factor for cancers [20,21]. Recently, several publications emphasized the importance of these POPs’ chemical properties regarding cancer development [20,21,22,23]. These compounds were extensively employed for industrial and consumer applications, although concerns about their toxicity and tendency to bioaccumulate in lipids led to global restrictions in manufacture over the past decades. POPs represent several families of chemicals, including dioxin-like substances, such as polychlorinated dioxins/furans (PCDD/PCDF), polychloro-biphenyls (PCBs), and polycyclic aromatic hydrocarbons (PAH, such as benzo[α]pyrene), that are chemically stable.

A few epidemiological studies reported a moderate increase in mortality from stomach cancers resulting from the ‘Seveso accident’, one of the best documented industrial accidents involving dioxin exposure [24,25,26]. Likewise, after the Yucheng accident in Central Taiwan, the cancer mortality rate was increased and associated with the accidental exposure to PCBs and PCDFs due to ingestion of contaminated cooking oil [27]. The incidence of gastric cancer was also increased among rubber tire workers [28] and residents living around a PCBs production site. Despite these accidental or occupational cases, to date, no study has investigated the selection of POPs that might play a role in gastric cancer, as well as the mechanisms involved.

POPs persist over long periods in the environment and contaminate drinking water and food [29]. They may enter the body through food and air inhalation, and, because they are highly lipophilic, they can accumulate in tissue with a high fat content [30,31]. Exposure results in a variety of toxic effects in experimental animals and human, including endocrine-disrupting effects and immunologic and carcinogenic changes [23,30,32,33]. The PCDD/PCDF concept is perhaps the more commonly used approach for human risk characterization and management [30]. Especially, the biology potency of dioxin-like substances refers to the most toxic dioxin (TCDD), to which a toxic equivalency factor (TEF) of 1 is assigned [30]. The concentration of POPs in adipose tissue is expressed in toxic equivalent units (TEQ), which is the product of the concentration of an individual compound and its corresponding TEF (TEQ = concentration × TEF).

The omentum is a poorly studied adipose tissue that covers the colon, small bowel, and ovaries in the abdominal cavity [34,35]. The greater omentum is typically resected from surgery of any gastric cancer. Different types of surgery, such as partial or total gastrectomy, and variable extents of lymph node resection are practiced. In all surgical procedures that include partial or total resection of the greater omentum, surgical specimens become available for analysis.

The aim of this pilot study was to conduct a preliminary exploration about the nature and concentration profiles of a large list of POPs in omental adipose tissue from patients with and without diffuse-gastric cancers. Because of the exploratory procedure include in a pilot study, we decided to evaluate others cancers for which an omentum resection is necessary. Because POPs exposition is common in our environment, definition of a control group is debating, considered as a patient non-affected by a cancer, or patient affected by a cancer for another primary cancer. This preliminary information becomes of critical relevance to design large-scale studies, providing essential knowledge for hypothesis generation, to prioritize chemical congeners and analytical methodologies, biological collection strategies, or depicting background exposure distributions for sample size calculations.

## 2. Materials and Methods

### 2.1. Patients and Study Design

Thirty-two patients were enrolled from 2013 to 2015 in the Department of Digestive and Oncology Surgery at Lariboisière Hospital, Paris (France), including 14 patients with an independent cell adenocarcinoma (diffuse-GC), 10 patients with other cancers (ovary or colon) that had metastasized in the peritoneal cavity, and 8 patients operated for non-cancer disease, as wound abdominal surgery. If patients met the eligibility criteria, they were completely informed about the research protocol and gave informed consent to participate in the study, which was approved by the ‘Comité de protection des personnes’ in 2013 in France (French equivalent of an Institutional Review Board, IRB). The protocol of this study was approved by the Bioethics Committee of the GNEDS (‘Groupe Nantais d’Ethique dans le Domaine de la Santé’). The greater omentum is typically resected during surgery for gastric, colon, and ovary cancer. In case of gastrectomy, a systematic sample (one centimeter) of the greater omentum without a macroscopic tumor implant or lymph node was resected. In case of other peritoneal extended cancers (ovary, colon, peritoneal pseudomyxoma), the cytoreductive surgery for carcinomatosis also includes the resection of the greater omentum. All diagnoses were confirmed by histopathology, and gastric cancers were classified as late stages III or IV according to the Lauren classification [3]. One patient was the source of 4 specimens for analysis of intra-person variability of samples. Control individuals were operated for hernia or surgical hernia. After informed consent, about 10 g of omental fat was removed from the abdominal cavity during surgery or under local anesthesia. The specimen was frozen (−80 °C) for later chemical analysis. All patient data were anonymized with a unique id-code that remained blind for its status as a case or control during the chemical analysis. Detailed information, such as anthropometric variables (age, sex, body mass index), and other factors potentially associated with exposure to POPs was recorded into a database. These variables had been used prior in the toxicological literature to adjust the results of POPs concentration.

### 2.2. Chemical Analysis

The methodologies applied to isolate, detect, and quantify the targeted POPs, including 17 dioxins (PCDD/F), 18 PCBs (dioxin-like PCBs + non-dioxin-like PCBs), 8 polybromodiphenylethers (PBDE), 6 polybromobiphenyls (PBBS), hexabromocyclododecanes (HBCDs), and 30 organochlorine pesticides (OCs), have been described previously [32,36,37]. Briefly, 13C-labeled congeners were added to each sample for quantification according to the isotopic dilution method. Lipids were extracted from adipose tissue samples under high pressure and temperature (ASE Dionex, Sunnyvale, CA, USA). The resulting extracts were dried and weighed to measure fat content using the gravimetric method. Gel permeation chromatography was used for isolating OCs, while three purification steps using successively acid silica, florisil, and celite/carbon columns were applied for other targeted substances. PCDD/F, PCB, PBDE, PBB, and OC measurements were performed by gas chromatography (Agilent 7890A, Nantes, France) coupled with high-resolution mass spectrometry (GC-HRMS) on electromagnetic sector instruments (JEOL MS 700D or 800D, Nantes, France), operating at a resolution of 10,000 and in the single ion monitoring (SIM) acquisition mode [32]. HBCD isomers were quantified using liquid chromatography coupled with tandem mass spectrometry (LC-MS/MS) on a triple quadrupole instrument (Agilent 6410). All data were generated in a manner blind to the control or case status of individuals. QA/QC procedures included systematic analysis of negative (blank, *n* = 3) and positive (standard reference material, *n* = 1) control samples in each batch of analyzed samples, and several inter-laboratory assays that were realized both nationally and internationally confirmed the robustness and accuracy of the protocols. All analyses were developed with performant methods and according to current European criteria in the field of routine control of foodstuffs of animal origin (EU, 2014). The analyses have been conducted in an ISO 17025:2005 accredited laboratory. Recoveries were in the 80–120% range, and the method’s extended uncertainty was lower than 20%. The omental content of POPs was expressed on lipid basis content as previously reported, measuring the fat content in adipose tissue gravimetrically.

### 2.3. Statistical Analysis

Distributions of participants’ characteristics and concentration data of POPs in adipose tissue were summarized as medians and interquartile ranges. The distribution of continuous variables and POP concentrations of cases were compared with controls using the Mann–Whitney *U* test. Chemicals with detection rates below 50% were excluded, and LOD/2 was attributed to non-detected samples. Correlation between POPs was evaluated using Spearman’s rank correlation coefficient and displayed with heatmaps. A preliminary exploratory analysis to identify the most relevant POPs contributing to the discrimination between groups was conducted with sparse partial least square analysis (sPLS-A) [38]. All statistical analyses were conducted with R software v.3.5.0 (Free Software Foundation)

## 3. Results

### 3.1. Demographic Characteristics

All patients (*n* = 32, 16 men and 16 women) were Caucasian and recruited in Europe (France). The clinical characteristics of case and control individuals in terms of age and body mass index (BMI), major factors determining the internal content of lipophilic POPs were similar among groups (Table 1). Patients with cancer had diffuse-GC or other tumors that metastasize in the peritoneal cavity.

### 3.2. Distribution of Persistent Organic Pollutants in Omental Fat

Summary distributions of POPs levels determined in omental fat tissue are displayed in Table 2, Table 3, Table 4 and Table 5. Detection frequencies were very high for PCDD/Fs, PCBs, PBDEs, or PBB153. Lower rates of detection were achieved for organochlorine pesticides, especially for endrine, aldrine, heptachlor, trans-heptachlor epoxyde, α-chlordane, γ-chlordane, α-endosulfan, β-endosulfan, endosulfan sulfate, methoxychlor, o,p-DDE, o,p-DDD, with detection frequencies ranging from 3 to 41%, thus being excluded from the summary table. None of the 20 screened PAHs were detected in omental adipose tissue.

The total concentrations of PCDDs, as normalized by lipid content, in omentum adipose tissue was significantly higher in patients with diffuse-GC (median = 166 pg/g lipid, 2.45×, *p* = 0.005) compared to the control group (median = 67.5 pg/g lipid) (Table 2). The toxic equivalence (WHO-TEQ-2005) of PCDD/PCDF in omental tissue in the two groups was 14.1 (9–20.4) for diffuse-GC versus 9.5 (7.2–11.7) for the control (*p* = 0.15) (Table 2). Statistically significantly higher levels were mainly observed for OCDD (112.8 pg/g, 2.5×, *p =* 0.003), followed by 1.2.3.6.7.8 HxCDD (19.3 pg/g, 2×, *p =* 0.035) and 1.2.3.4.6.7.8 HpCDD (18.2 pg/g, 3×, *p* = 0.029) (Table 2). In contrast, concentrations of total PCBs (including dioxin-like (dl) and non-dl PCB), PBDEs, or organochlorine pesticides, were similar between patients with diffuse-GC and controls (Table 3 and Table 4). The main fraction of dioxin-like PCBs was not significantly different between diffuse-GC (median 54,709 pg/g) and controls, and represent a WHO-TEQ (TEF2005) equivalent of 9.5 versus 9.4 (*p* = 0.53) (Table 3). Moreover, a polybromodiphenyl-ether (PBDE209) and one polybromobiphenyl (PBB153) were significantly higher in patients with diffuse-GC in comparison to the control group (*p* = 0.005 and *p* = 0.042, respectively) (Table 4). A few organochlorine pesticides, such as p,p-DDE and b-hexachlorobenzene, were detected in the human omentum, although not statistically different in patients with diffuse-GC vs controls at *p* < 0.05 (Table 5). In addition to analysis performed in patients with diffuse-GC, the concentrations of POPs were analyzed in another group of patients with peritoneal metastatic cancers (including ovarian and colon cancers, as defined by ‘other tumors’). A few PCBs (PCB105, PCB123, and PCB138) were significantly higher in the group ‘other tumors’ as compared to controls (Table 3).

### 3.3. An Exploratory Mixture Analysis

The correlation between POPs, mostly detected in omental fat (PCDD/Fs, PCBs, PBDEs, PBB153), is displayed in Figure 1. The heatmap illustrates the Spearman’s rank correlation coefficient between two congeners. The results showed a very strong correlation between most PCDD/Fs, PCBs and few PBDEs (e.g., PBDE 153 and 209) and PBB153. Some congeners depicted weak or inverse correlation pattern with the rest of POPs, including PBDE 28, 47, 99 or PCB52, 101 (Figure 1).

The principal component analysis further indicated that the first and second components accounted for most of model variance (54% and 12%, respectively, for Dim1 and Dim2) (Figure 2A). The individuals displayed in the first two principal components were not homogeneously distributed (Figure 2A). The groups of “diffuse-GC” and “control” appeared to be the most disassociated. Moreover, the sPLS-DA (Figure 2B) showed that -the OCDD and Sum of PCDDs (more abundant in the “diffuse-GC” group), –1.2.3.4.7.8-HxCDD (more abundant in the “other metastatic cancer” group), and –PCB52 (more abundant in the “control” group), were the most discriminant variables.

## 4. Discussion

This pilot study explored, for the first time, the concentration of almost one hundred persistent organic pollutants (POPs) in the human omental tissue from French patients with diffuse-gastric adenocarcinoma, other cancers, and of a control group without the disease. The data reveal a widespread presence of low amounts of most POPs (PCDDs/Fs, PCBs, and brominated flame retardants) in the human omentum with minor presence of some organochlorine pesticides and PAHs at the low analyzed levels.

The amounts of POPs in the human omentum still remains scarcely explored compared to other adipose tissue locations, such as peripheral fat. Whereas some studies have reported high correlations between concentrations levels of POPs across different fat pads (i.e., sub-cutaneous, visceral), other studies have found substantial differences and low correlations [31,39,40,41]. Methodological limitations on the biological collection omental fat may prevent the use of this matrix in research. The feasible application of omental fat in research on the associations between POPs and endometriosis has been conducted within the innovative framework of the ENDO study where a population-based cohort was matched to a clinical one, favoring a valuable biocollection [42]. In a previous study conducted within this framework, we found comparable concentrations and high bivariate correlations for most POPs between fat pads (sub-cutaneous vs omental) [32]. The greater omentum (i.e., Epiplon) represents a specific large flat adipose tissue that covers digestive organs and moves around the peritoneal cavity. The omentum is a common site for advanced gastro-intestinal cancer and is usually affected by peritoneal metastasis [43]. Thus, the specific concentrations of POPs in the greater omentum might directly reflect the target microenvironment, highlighting its valuable application in research in gastro-intestinal cancers.

In the present study, the analytical method was selective and sensitive enough to have high detection frequency. The concentrations of POPs reported in our study are similar to those previously reported in France, indicating a background exposure profile comparable with that of the general population [32]. For instance, the median concentration of TCDD ranged between 1.2 and 1.3 pg/g in the present study, whereas median concentrations s ranging between 0.7 and 1.4 pg/g were reported in adipose tissue from French women recruited during a similar period [23,32]. Similar comparable concentrations and detection frequencies were also found for PCBs, PBDEs, or most organochlorine pesticides. Noticeably higher concentrations (about 3-fold) were observed in the present study for hexachlorobenzene, b-HCH, trans-nanochlor, or p,p’-DDE compared with the endometriosis study [32]. Similar bivariate correlation signatures have been reported before for the list of analyzed POPs. Strong positive correlations between PCDD/Fs and PCBs and negative correlations were found for PBDE28 or 47 [38].

The goals and design of the present study do not allow for addressing more complex questions, such as the potential associations between POPs and cancer risk. Nonetheless, the preliminary analysis suggested some statistical differences among groups. For instance, significantly higher levels of PCDD/F were observed in diffuse-GC as compared to the control group, mainly due to OCDD (median >112.8 pg/g omental adipose tissue, *p = 0.003*), followed by 1.2.3.4.7.8 HpCDD and 1.2.3.4.7.8 HxCDD (Table 2). Other chemicals were also found in larger concentrations in the group of “other cancers”, including 1.2.3.4.7.8-HxCDF, 1.2.3.6.7.8-HxCDF, OMS-TEQ dl-PCBs, Total-TEQ, or the sum of 6 non-dioxin-like PCBs, among others (Table 2 and Table 3). These results are not conclusive themselves due to the limited sample size of patients and lack of adjustment for confounding variables. However, a larger study that include both diffuse-GC and other aggressive tumors that metastasize in the omentum (*n* = 32) showed significant adjusted OR for WHO-TEQ PCDD/PCDF and dlPCB (TEF2005) when compared to controls (patients without tumors) (*p < 0.05*). Crude ORs, adjusted OR, and corresponding 95% confidence intervals were estimated in an associated analysis (additional Appendix A). Further research with large scale studies are needed to evaluate these associations, all influenced by geographic patients’ locations.

Several epidemiological studies have reported an increased mortality rate from cancer, including stomach cancer after accidental or occupational exposure to POPs, such as dioxin, and in cohorts of chemical manufacturing workers producing PCDD/PCDF, PCBs, and PAHs [24,25,28,44,45,46]. The associations between PCDD/PCDF and diffuse-GC are supported by several studies using experimental models. For instance, mice treated with TCDD or PCDD/HCDF all had hyperplasia in the fundus of glandular stomach [47]. Sub-chronic exposure to OCDD appears to cause effects similar to those observed following exposure to low levels of TCDD in a rat model [48,49], although with less potency in human cells [49]. TCDD, the most toxic compound among PCDD/F family, also triggers invasive processes in human gastric tumor cell lines [50]. The proposed molecular mechanism of PCDD/PCDFs envisages the binding of the dioxin to the aryl hydrocarbon receptor, which is a transcriptional regulator of cell growth, migration, and invasion of cancer cells, as well as inflammation and immune suppression in several cancers [51,52,53]. Besides this classical pathway, a number of papers are now dealing with the role of epigenetic mechanisms in response to environmental xenobiotics, such as TCDD and OCDD, or PCB compounds [54,55,56,57].

## 5. Conclusions

To sum up, the present pilot study provides, for the first time, essential information regarding POPs levels in the omental fat, which is a relatively understudied fat depot in terms of POPs load. The study confirms the widespread presence of most relevant PCDD/Fs, PCBs, PBDEs, and organochlorine pesticides, despite the fact that they were banned in the 1970s. The information provided by the present study will support the prioritization of chemicals, biological matrix sampling strategy, and sample size calculations for future studies in patient’s cohorts with gastric cancers. Capturing the complexity of chemicals mixtures should remain a priority over the analysis of single chemicals as confirmed in the present study. Environmental chemicals may be involved in the carcinogenic processes through multiple direct and indirect pathways, including the alteration of the poorly understood adipocyte-tumoral cell cross-talk from cases with gastric cancers. Omental fat represents a target tissue of critical relevance on the pathogenesis of gastric cancer that should be seriously considered in the research of environmental risk factors including persistent pollutants.

## Figures and Tables

**Figure 1 cancers-13-04874-f001:**
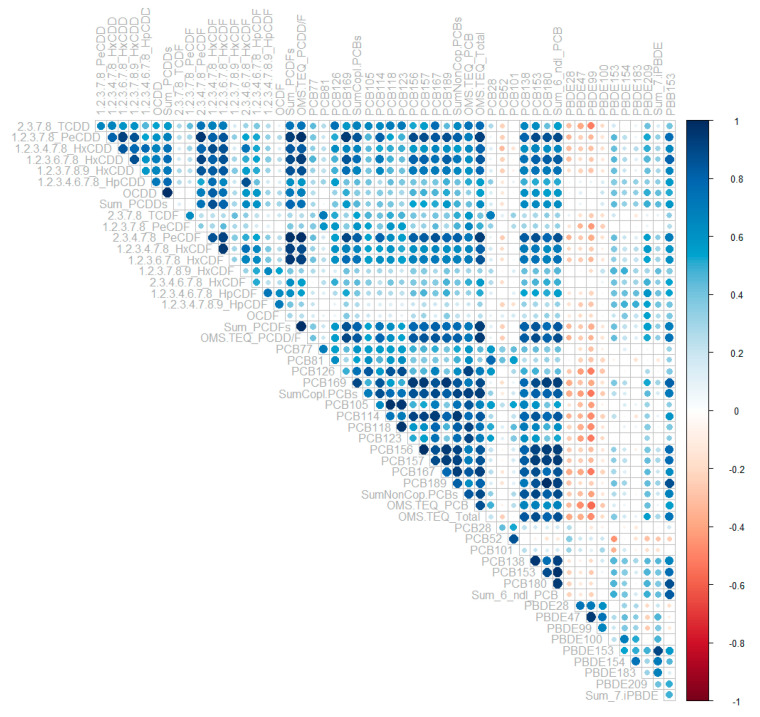
Heatmap displaying the Spearman’s rank correlation coefficients between main congeners of PCDD/Fs, PCBs, PBDEs, and PBB153. The color intensity relates to the correlation coefficient magnitude, and the color to the direction (blue, positive associations; red, negative associations). Of note, and unsurprisingly, PCDD correlates with each other, as does PCDF, in our study; certain ‘volatile’ POPs, such as PBDE28 (and PBDE47), PCB28, and PCB52, known to be metabolized, are negatively correlated to the lipophilic POPs.

**Figure 2 cancers-13-04874-f002:**
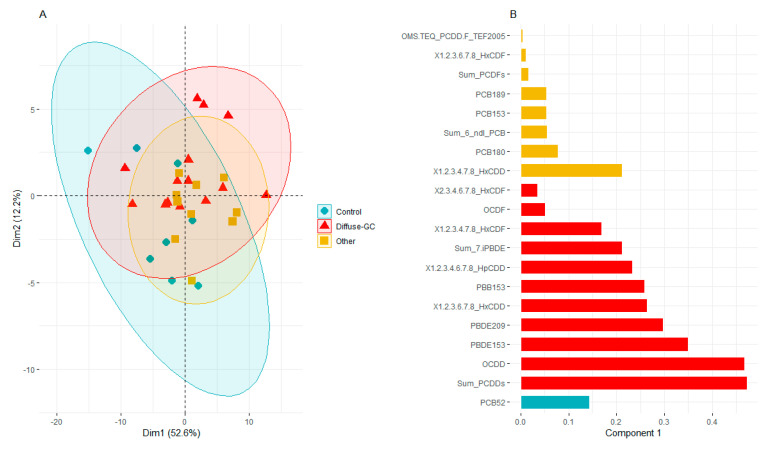
(**A**) The projection of individuals in the first two dimensions from the principal component analysis. (**B**) The contribution of each pollutant to the discrimination of three main groups (“diffuse-GC”; “other metastatic cancers”; and “control” groups, *n* = 32) is represented in the bar plot of loading weights computed from the first dimension from the sparse partial least square discriminant analysis. The color key in (**B**) indicates the group where the chemicals are more abundant.

**Table 1 cancers-13-04874-t001:** Summary characteristics of patients (median and interquartile range) without cancer (controls), with diffuse-gastric adenocarcinoma, with other tumors that metastasis in the peritoneal cavity. Age and BMI of case groups were compared with controls using Mann–Whitney *U* test without statistical differences.

Patients	Controls (*n* = 8)	Diffuse-GC (*n* = 14)	Other Tumors (*n* = 10)
**Age (years)**	63.5 (50.2–69.2)	55.0 (48.5–60.5)	66.0 (60.2–69.5)
**BMI (kg/m^2^)**	26.9 (24.3–29.5)	23.6 (21.1–27.2)	28.2 (23.9–31.1)
**Sex (% female)**	50%	57%	40%

**Table 2 cancers-13-04874-t002:** Summary of polychlorinated dioxins/furans (PCDDs/PCDFs) distributions and detection frequencies (DF) measured in omental fat from cases of diffuse-gastric cancer, other cancers with peritoneal metastasis, and non-cancer as control patients. Results are represented by median (interquartile range) (pg/g lipid). The case groups were compared to the control group with the Mann–Whitney *U* test (* *p* < 0.05; ** *p* < 0.01).

PCDDs/PCDFs	Controls (*n* = 8)	Diffuse-GC (*n* = 14)	*p*	Other Cancers (*n* = 10)	*p*
2.3.7.8-TCDD	1.2 (0.6–1.2)	1.3 (1.0–1.8)	0.19	1.3 (1.0–2.3)	0.20
1.2.3.7.8-PeCDD	3.6 (2.9–4.3)	5.1 (3.4–7.3)	0.27	4.8 (3.7–7.7)	0.15
**1.2.3.4.7.8-HxCDD**	1.1 (1.0–1.7)	2.3 (1.4–3.8)	**0.03**	2.2 (1.6–5.3)	**0.02**
**1.2.3.6.7.8-HxCDD**	9.7 (7.8–12.5)	19.3 (10.4–29.3)	**0.03**	16.2 (12.6–27.3)	**0.03**
1.2.3.7.8.9-HxCDD	1.4 (0.9–1.8)	1.9 (1.0–2.9)	0.27	1.4 (1.1–2.9)	0.32
**1.2.3.4.6.7.8-HpCDD**	5.8 (5.1–7.5)	18.2 (9.3–25.3)	**0.03**	15.6 (8.3–21.7)	0.055
**OCDD**	44.5 (36.1–56.3)	112.8 (74.6–259)	**0.003**	63.1 (57.6–95.5)	0.08
**Sum PCDDs**	67.5 (54.7–94.4)	165.7 (105.3–319)	**0.005**	102.3 (86.8–138)	**0.04**
2.3.7.8-TCDF	0.3 (0.3–0.9)	0.3 (0.2–0.5)	0.87	0.5 (0.2–0.6)	0.97
1.2.3.7.8-PeCDF	0.2 (0.1–0.4)	0.1 (0.1–0.2)	0.66	0.3 (0.2–0.4)	0.27
2.3.4.7.8-PeCDF	9.2 (6.9–12.0)	15.0 (8.2–20.4)	0.13	12.4 (10.2–19.2)	0.08
1.2.3.4.7.8-HxCDF	1.7 (1.5–2.3)	2.9 (2.1–4.4)	0.11	2.6 (2.2–3.5)	**0.04**
1.2.3.6.7.8-HxCDF	2.0 (1.6–2.8)	3.1 (2.1–5.1)	0.15	3.2 (2.5–4.7)	**0.04**
1.2.3.7.8.9-HxCDF	0.1 (0.1–0.2)	0.1 (0.1–0.2)	0.92	0.1 (0.1–0.2)	0.97
2.3.4.6.7.8-HxCDF	0.7 (0.5–0.9)	1.1 (0.6–1.7)	0.11	1.0 (0.6–1.5)	0.12
1.2.3.4.6.7.8-HpCDF	1.1 (1.0–1.2)	1.4 (0.9–3.0)	0.24	1.4 (1.0–1.8)	0.17
1.2.3.4.7.8.9-HpCDF	0.1 (0.1–0.1)	0.2 (0.1–0.3)	0.15	0.1 (0.1–0.2)	0.20
OCDF	0.2 (0.2–0.3)	0.3 (0.3–0.5)	0.055	0.3 (0.2–0.4)	0.24
Sum PCDFs	17.7 (12.9–21.0)	23.4 (15.8–34.2)	0.11	21.4 (18.5–35.8)	0.10
WHO-TEQ PCDD/F (TEF2005)	9.5 (7.2–11.7)	14.1 (9.0–20.4)	0.15	12.6 (10.3–19.6)	0.07

In bold, significance represented by *p* < 0.05 for comparisons between cases and controls.

**Table 3 cancers-13-04874-t003:** Summary of polychlorobiphenyls (PCBs) distributions and detection frequencies (DF) measured in omental fat from cases of diffuse-gastric cancer, other cancers with peritoneal metastasis, and non-cancer as control patients. Results were represented by median (interquartile range) (pg/g lipid). The case groups were compared to the control group using Mann–Whitney *U* test (*p* < 0.05).

PCBs	Controls (*n* = 8)	Diffuse-GC (*n* = 14)	*p*	Other Tumors (*n* = 10)	*p*
PCB 77	2.3 (1.6–4.2)	1.9 (1.5–3.3)	0.87	3.1 (2.3–4.3)	0.57
PCB 81	1.2 (0.8–3.3)	1.0 (0.7–2.3)	0.57	2.4 (1.6–3.8)	0.17
PCB 126	45.2 (30.9–61.0)	42.0 (24.5–92.0)	0.82	79.1 (56.8–97.9)	*0.055*
PCB 169	77.9 (41.0–111.2)	117.1 (73.6–142.9)	0.27	111.5 (89.6–193.1)	0.10
Sum Copl. PCBs	129.1 (100.8–180.5)	150.7 (121.3–248.4)	0.44	188.9 (160.5–282.8)	*0.055*
**PCB 105**	3217 (1603.7–4005.3)	3541 (1928–5539.7)	0.53	**6300** (3596.8–12,063.8)	**0.03**
PCB 114	1085.4 (505.7–2353.2)	1559.9 (1234.8–2009.8)	0.62	2389.8 (1317.8–3719.1)	0.15
PCB 118	15,516 (7550–17,555)	17,402 (962–24,966)	0.40	30,968 (17,172–56,890)	0.055
**PCB 123**	133.0 (60.2–154.9)	139.0 (65.0–283.7)	0.57	262.9 (194.8–504.4)	**0.02**
PCB 156	16,529(5050–24,869)	24,763(15,818–32,199)	0.37	24,793 (17,099–41,779)	0.20
PCB 157	2745.3 (1115.9–4501.2)	4583.6 (3252–5519)	0.21	5136.4 (3952.7–7858.7)	0.07
PCB 167	2336.1 (1344.8–3962.3)	3360.5 (2843–5490)	0.37	4579.9 (3248.6–6311.7)	0.08
PCB 189	2457.6 (684.1–3209.8)	4087.6 (2275–5095)	0.09	2949.6 (2741.8–5980.2)	0.12
Sum Non Cop. PCBs	39,659 (26,274–66,540)	54,709 (48,975–70,840)	0.27	78,909 (50,996–146,650)	*0.055*
**WHO-TEQ dl-PCB** **(TEF 2005)**	9.4 (6.0–10.7)	9.5 (7.3–16.2)	0.53	13.3 (11.8–16.6)	**0.02**
**TOTAL-TEQ** **(TEF 2005)**	19.6 (12.5–21.3)	22.6 (18.3–36.8)	0.15	24.7 (23.0–35.9)	**0.03**
PCB 28	1.3 (1.1–1.9)	1.3 (0.8–1.7)	0.71	1.5 (0.9–2.9)	0.83
PCB 52	0.5 (0.2–0.6)	0.2 (0.1–0.3)	*0.03*	0.2 (0.2–0.6)	0.27
PCB 101	0.7 (0.4–1.4)	0.5 (0.2–0.7)	0.21	0.6 (0.5–1.1)	0.83
**PCB 138**	34.7 (31.7–92.6)	78.6 (52.8–113.2)	0.19	91.6 (66.6–137.5)	**0.04**
PCB 153	97.0 (66.7–198.1)	194.0 (136.9–261.3)	0.21	238.5 (147.5–254.7)	0.10
PCB 180	141.4 (50.7–179.9)	233.2 (138.0–315.9)	0.11	190.0 (177.6–315.3)	*0.055*
**Sum 6 ndl-PCB**	288.3 (156.7–448.3)	495.4 (345.8–666.5)	0.19	574.3 (386.9–757.6)	**0.04**

In bold, significance represented by *p* < 0.05 for comparison with control.

**Table 4 cancers-13-04874-t004:** Summary of polybromodiphenylethers (PBDE) and polybromobiphenyls (PBB) distributions and detection frequencies (DF) measured in omental fat from cases of diffuse-gastric cancer, other cancers with peritoneal metastasis, and non-cancer as control patients. Results were represented by median (ng/g lipid (interquartile range). The case groups were compared to the control group with Mann–Whitney *U* test (*p* < 0.05).

PBDE	Controls (*n* = 8)	Diffuse-GC (*n* = 14)	*p*	Other Tumors (*n* = 10)	*p*
PBDE 28	0.016 (0.014–0.028)	0.016 (0.012–0.020)	0.40	0.009 (0.005–0.028)	0.24
PBDE 47	0.151 (0.106–0.245)	0.165 (0.048–0.330)	0.82	0.075 (0.033–0.212)	0.27
PBDE 99	0.038 (0.026–0.065)	0.047 (0.012–0.066)	0.97	0.022 (0.011–0.064)	0.46
PBDE 100	0.074 (0.053–0.097)	0.076 (0.052–0.138)	0.71	0.067 (0.051–0.083)	0.70
PBDE 153	0.847 (0.636–2.613)	2.141 (1.682–2.598)	0.09	1.831 (1.638–2.070)	0.32
PBDE 154	0.015 (0.010–0.020)	0.019 (0.013–0.031)	0.40	0.021 (0.016–0.028)	0.32
PBDE 183	0.178 (0.068–0.321)	0.219 (0.156–0.389)	0.37	0.289 (0.228–0.366)	0.10
**PBDE 209**	1.860 (1.260–2.786)	5.014 (3.112–9.353)	**0.005**	2.755 (2.336–6.320)	0.10
Sum 7 i PBDE	1.346 (0.963–3.372)	2.627 (2.221–3.591)	0.09	2.392 (2.213–2.976)	0.27
**PBB 153**	0.506 (0.340–0.574)	1.069 (0.523–1.489)	**0.04**	0.766 (0.673–1.163)	**0.006**

In bold, significance represented by *p* < 0.05 for comparison with control.

**Table 5 cancers-13-04874-t005:** Summary of organochlorinated pesticides (OCPs)distributions and detection frequencies (DF) measured in omental fat from cases of diffuse-gastric cancer, other cancers with peritoneal metastasis, and non-cancer as control patients. Results were represented by median (interquartile range) (ng/g lipid). The case groups were compared to the control group using Mann–Whitney *U* test (*p* < 0.05).

OCPs	Controls (*n* = 8)	Diffuse-GC (*n* = 14)	*p*	Other Tumors (*n* = 10)	*p*
Missing	1 (12.5%)	3 (21.4%)		1 (10.0%)	
Hexachlorobenzene	24.8 (16.1–28.7)	28.8 (17.4–38.6)	0.54	24.5 (20.9–30.2)	0.54
Pentachlorobenzene	0.3 (0.3–0.4)	0.4 (0.3–0.5)	0.25	0.4 (0.3–0.4)	0.61
aHCH	1.4 (1.3–1.7)	1.7 (1.4–2.1)	0.29	1.7 (1.7–2.3)	**0.04**
bHCH	87.8 (40.8–123.1)	54.9 (34.7–101.8)	0.54	75.1 (55.1–119.2)	0.61
gHCH	3.6 (2.9–4.2)	4.2 (3.5–5.1)	0.33	4.7 (4.0–5.5)	0.07
dHCH	1.1 (0.9–1.4)	1.0 (0.9–1.2)	0.43	1.3 (1.0–1.4)	0.76
a-Chlordane *	0.2 (0.1–0.3)	0.3 (0.1–0.6)	0.48	0.2 (0.1–0.3)	0.76
g-Chlordane *	0.3 (0.2–0.5)	0.5 (0.1–0.8)	0.29	0.5 (0.3–0.7)	0.3
Cis_nonachlore	2.4 (1.4–3.3)	2.2 (1.8–2.9)	0.93	4.4 (2.8–5.0)	0.09
Trans_nonachlore	15.5 (8.8–17.8)	15.3 (12.3–17.2)	0.79	24.5 (21.4–26.7)	**0.01**
Oxychlordane	16.0 (10.7–21.9)	17.2 (11.0–25.7)	0.72	28.4 (19.4–34.4)	0.05
Heptachlore *	0.0 (0.0–0.1)	0.0 (0.0–0.1)	0.72	0.0 (0.0–0.1)	1
Cis_heptachlore_epoxyde	11.3 (6.6–12.3)	9.3 (7.0–16.4)	0.79	17.4 (14.8–25.1)	**0.04**
Trans_heptachlore epoxyde *	0.5 (0.3–0.6)	0.7 (0.1–1.2)	0.29	0.4 (0.1–0.7)	0.68
Aldrine *	0.1 (0.0–0.1)	0.0 (0.0–0.2)	0.48	0.0 (0.0–0.1)	1
Dieldrine	11.5 (8.7–17.0)	9.8 (6.6–13.4)	0.66	14.0 (10.1–18.4)	0.68
Endrine *	0.1 (0.1–0.2)	0.2 (0.1–0.3)	0.6	0.2 (0.1–0.2)	0.61
a-Endosulfan *	0.8 (0.2–0.9)	0.4 (0.1–1.0)	0.79	0.6 (0.1–0.6)	0.76
b-Endosulfan *	0.6 (0.3–0.8)	0.5 (0.2–1.1)	0.93	0.5 (0.2–1.2)	0.84
Endosulfan_sulfate *	0.2 (0.1–0.3)	0.2 (0.1–0.3)	0.66	0.3 (0.2–0.3)	0.61
p,p-DDT	6.3 (5.4–15.3)	3.7 (2.7–6.5)	0.25	3.2 (2.7–9.5)	0.41
o,p-DDT	1.3 (0.9–1.9)	0.7 (0.3–1.8)	0.54	0.8 (0.4–1.0)	0.21
p,p-DDE	255 (102–799)	168 (127–279)	0.72	251(148–521)	1
o,p-DDE *	0.2 (0.1–0.4)	0.2 (0.1–0.8)	0.86	0.1 (0.1–0.2)	0.14
p,p-DDD	1.9 (1.5–3.8)	0.7 (0.6–1.2)	0.07	1.2 (0.7–2.2)	0.41
o,p-DDD *	0.2 (0.1–0.3)	0.1 (0.1–0.4)	1	0.2 (0.1–0.3)	0.92
Methoxychlore *	0.5 (0.2–0.6)	0.3 (0.1–1.1)	0.79	0.3 (0.1–0.4)	0.35
Mirex	3.7 (1.0–5.1)	3.3 (1.9–7.2)	0.6	4.5 (3.4–8.0)	0.21

* Note: >50% pollutants were not detected or at the limit of detection. In bold, significance represented by *p* < 0.05 for comparison with control.

## Data Availability

The data presented in this study are available on request from the corresponding author.

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
