# Peer review of "Detection of Persistent Organic Pollutants in Omental Adipose Tissue from Patients with Diffuse-Gastric Cancer: A Pilot Study"

_cancers, 2021, doi:10.3390/cancers13194874_

Round 1

Reviewer 1 Report

I read with great interest the manuscript entitled “Detection of persistent organic pollutants in omental adipose tissue from patients with diffuse-gastric cancer: a pilot study”. While the concept of searching a large panel of 96 POPs in the momentum for gastric cancer is more than attractive, the current study is not well structured.

The authors do mention this is a pilot study, however this fact does not cover all the drawbacks.

First of all, the title mentions that the objective is to detect persistent organic pollutants in omental adipose tissue from patients with diffuse gastric cancer. The inhomogeneity of the patients included (14 with gastric cancer, 10 patients with ovary and colon and 8 other patients which underwent surgery for other reasons) surely does not stand as a strength in this study. It is not clear if the study compares gastric cancer vs other cancers vs no cancer. Most of the discussions are about gastric cancer vs controls. The authors do comment of another study, even if it is mentioned with (not shown – line 285) which compares cancer patients (gastric included) with controls. Thus, for the number of patients included, a similar protocol might be more effective, since other POPs were found higher in other cancer when compared to controls.

On the other hand, major limitations were not fully discussed, and also only demographic data regarding age, BMI and sex were considered relevant to be included in the study.

What does 8 healthy volunteers mean? I find it unfortunate the statement “8 healthy volunteers”. Perhaps controls and cancer free patients, since they underwent surgery for other reasons. Moreover, at Line 105 authors mention that patients “were operated for hernia or surgical hernia”

Also, just a few other minor language inconsistancies:

Line 45 – has also been pointed as proposed as risk factors

Line 46 - several publications emphases

Line 73 – covers several digestive organs and ovaries

Line 76 – in all surgeries

Author Response

Response to Reviewer 1 Comments

Point 1

I read with great interest the manuscript entitled “Detection of persistent organic pollutants in omental adipose tissue from patients with diffuse-gastric cancer: a pilot study”. While the concept of searching a large panel of 96 POPs in the momentum for gastric cancer is more than attractive, the current study is not well structured.

We thanks the reviewer for the great interest and try to improve the manuscript.

The authors do mention this is a pilot study, however this fact does not cover all the drawbacks.

First of all, the title mentions that the objective is to detect persistent organic pollutants in omental adipose tissue from patients with diffuse gastric cancer. The inhomogeneity of the patients included (14 with gastric cancer, 10 patients with ovary and colon and 8 other patients which underwent surgery for other reasons) surely does not stand as a strength in this study.

It is not clear if the study compares gastric cancer vs other cancers vs no cancer. Most of the discussions are about gastric cancer vs controls. The authors do comment of another study, even if it is mentioned with (not shown – line 285) which compares cancer patients (gastric included) with controls. Thus, for the number of patients included, a similar protocol might be more effective, since other POPs were found higher in other cancer when compared to controls.

Response 1:

We have modified the manuscript to be more easy to explain and analyze and to compare differently the 3 groups as noted by the reviewer. For that we offer to the readers different analysis including additional data.

Of course some POPs could be more higher in others cancers when compared to diffuse gastric cancers or control regarding the multi causal origin of colon, ovarian or others cancers.

Sentence had been include to try to explain the decision of 3 groups in the introduction:

Because of the exploratory procedure include in a pilot study, we decided to evaluate others cancers for which an omentum resection is necessary. Because POPs exposition is common in our environment, definition of a control group is debating, considered as patient non affected by a cancer, or patient affected by a cancer for another primary cancer.

Table 1 had been simplified with only 3 group proposed.

Additional data have been proposed with comparison between the different groups.

Point 2 : On the other hand, major limitations were not fully discussed, and also only demographic data regarding age, BMI and sex were considered relevant to be included in the study.

Response 2:

In the literature these data are used to correct part of the results regarding the fact that BMI could modified the global POPs exposure. The rational is not so strong, so we decided to give these information if readers looking for that.

A sentence had been added in the method section : These variables had been used prior in the toxicological literature to adjust the results of POPs concentration.

Point 3 : What does 8 healthy volunteers mean? I find it unfortunate the statement “8 healthy volunteers”. Perhaps controls and cancer free patients, since they underwent surgery for other reasons. Moreover, at Line 105 authors mention that patients “were operated for hernia or surgical hernia”

Response 3:

The sentence had been clarified as required by the reviewer.

8 patients operated for non cancer disease, as wound abdominal surgery healthy volunteers.

Also, just a few other minor language inconsistancies:

Line 45 – has also been pointed as proposed as risk factors

Sentence had been changed as : Exposure to environmental persistent organic pollutants (POPs) has also been pointed as proposed as risk factors for cancers [20-21].

Line 46 - several publications emphases

Sentence had been changed as : Exposure to environmental persistent organic pollutants (POPs) has also been pointed as proposed as risk factors for cancers [20-21]. and Recently several publications emphases the importance of these POPs’s chemicals property regarding cancer development [20-23].

Line 73 – covers several digestive organs and ovaries

Sentence had been changes as : The omentum is a poorly studied adipose tissue that covers several digestive organs colon, small bowel and ovaries in the abdominal cavity [34, 35].

Line 76 – in all surgeries

Sentence had been changed as : In all surgeries surgical procedures that include partial or total resection of the greater omentum, surgical specimens become available for analysis.

Thank you for the review and interest .

Reviewer 2 Report

I think the authors have done a great job in this study. The findings are interesting and as the authors state are limited. 

The authors need to identify the sources of the toxins specific to gastric cancer and materials the patients potentially were exposed to. 

The other cancer column is a little confusing. I am unclear on the role of other cancers in this study. And the authors need to elaborate to the rational for including it. I think the authors would need to show differences between the GC and other cancers and the meaning of this and further illustrate why these differences are present or exclude. 

Author Response

Response to Reviewer 2 Comments

We thanks the reviewer for the great interest and try to improve the manuscript.

I think the authors have done a great job in this study. The findings are interesting and as the authors state are limited. 

The authors need to identify the sources of the toxins specific to gastric cancer and materials the patients potentially were exposed to. 

Patients could be exposed in different times and situations. As for other toxic, as exposure to asbestos may have occurred 25 years earlier, patients did not remember their life style or place they could be.

The other cancer column is a little confusing. I am unclear on the role of other cancers in this study. And the authors need to elaborate to the rational for including it. I think the authors would need to show differences between the GC and other cancers and the meaning of this and further illustrate why these differences are present or exclude.

Response:

We have modified the manuscript to be more easy to explain and analyze and to compare differently the 3 groups as noted by the reviewer. For that we offer to the readers different analysis including additional data.

Sentence had been include to try to explain the decision of 3 groups in the introduction, as asked by the reviewer:

Because of the exploratory procedure include in a pilot study, we decided to evaluate others cancers for which an omentum resection is necessary. Because POPs exposition is common in our environment, definition of a control group is debating, considered as patient non affected by a cancer, or patient affected by a cancer for another primary cancer.

Table 1 had been simplified with only 3 group proposed.

Additional data have been proposed with comparison between the different groups.

Thank you for the review and interest .

Round 2

Reviewer 1 Report

Thank you clarifying my concerns.